# LEARNING ROBUST VISUAL REPRESENTATIONS USING DATA AUGMENTATION INVARIANCE

## ABSTRACT

Deep convolutional neural networks trained for image object categorization have shown remarkable similarities with representations found across the primate ventral visual stream. Yet, artificial and biological networks still exhibit important differences. Here we investigate one such property: increasing invariance to identity-preserving image transformations found along the ventral stream. Despite theoretical evidence that invariance should emerge naturally from the optimization process, we present empirical evidence that the activations of convolutional neural networks trained for object categorization are not robust to identity-preserving image transformations commonly used in data augmentation. As a solution, we propose *data augmentation invariance*, an unsupervised learning objective which improves the robustness of the learned representations by promoting the similarity between the activations of augmented image samples. Our results show that this approach is a simple, yet effective and efficient (10 % increase in training time) way of increasing the invariance of the models while obtaining similar categorization performance.

**Keywords:** deep neural networks; visual cortex; invariance; data augmentation

## 1 INTRODUCTION

Deep artificial neural networks (DNNs) have borrowed much inspiration from neuroscience and are, at the same time, the current best model class for predicting neural responses across the visual system in the brain (Kietzmann et al., 2017; Kubilius et al., 2018). Yet, despite consensus about the benefits of a closer integration of deep learning and neuroscience (Bengio et al., 2015; Marblestone et al., 2016), important differences remain.

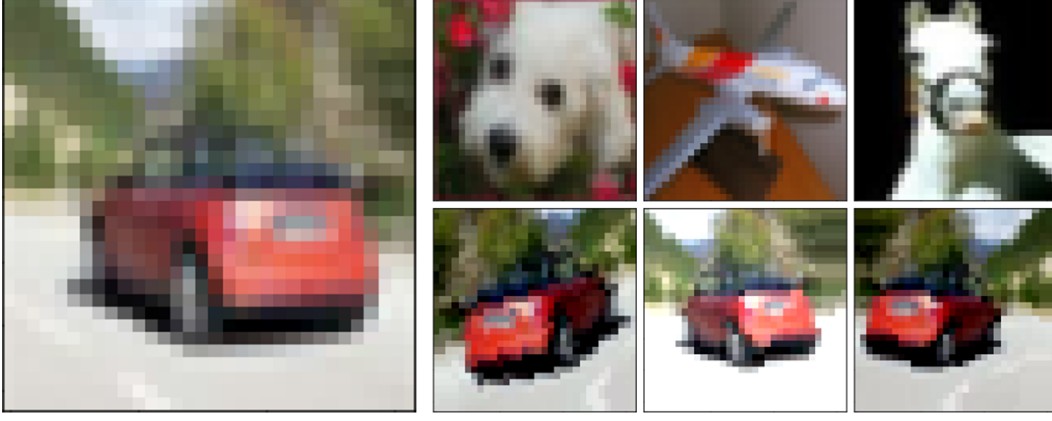

Figure 1: The top layer representations of the six images on the right learned by a prototypical CNN are equally (dis)similar to the reference image (left), even though the images at the bottom row are transformations of it.

Here, we investigate a representational property that is well established in the neuroscience literature on the primate visual system: the increasing robustness of neural responses to identity-preserving image transformations. While early areas of the ventral stream (V1-V2) are strongly affected by variation in e.g. object size, position or illumination, later levels of processing are increasingly robust to such changes, as measured first in single neurons of the inferior temporal (IT) cortex of macaques Booth & Rolls (1998) and then in humans' (Quiroga et al., 2005; Isik et al., 2013). The cascaded achievement of invariance to such identity-preserving transformations has been proposed as a key mechanism for robust object recognition (DiCarlo & Cox, 2007; Tacchetti et al., 2018).

Learning such invariant representations has been a desired objective since the early days of artificial neural networks (Simard et al., 1992). Accordingly, a myriad of techniques have been proposed to attempt to achieve tolerance to different types of transformations (see Cohen & Welling (2016) for a review). Interestingly, recent theoretical work (Achille & Soatto, 2018) has shown that invariance to "nuisance factors" should naturally emerge from the optimization process of deep models that minimize the information of the representations about the inputs, while retaining the minimum information about the target, as proposed by Tishby & Zaslavsky (2015) in the information bottleneck principle.

Nevertheless, DNNs are still not robust to identity-preserving transformations, including simple image translations (Zhang, 2019). A remarkable extreme example are adversarial attacks (Szegedy et al., 2013), in which small changes, imperceptible to the human brain, can alter the classification output of the network. Extending this line of research, we use the data augmentation framework (Hernández-García & König, 2018) to show that DNNs, despite being trained on augmented data, are not robust to such transformations and the learned representations are even less disentangled than in the input space.

This is likely related to the growing evidence that DNNs may exploit highly discriminative features that do not match human perception (Jo & Bengio, 2017; Ilyas et al., 2019; Wang et al., 2019). We postulate that this is due to the combination of their large capacity and the highly unconstrained learning setup of typical supervised deep models, and that incorporating elements from human visual perception and biological constraints can add a positive inductive bias that may yield better, more robust representations.

Hence, inspired by the increasing invariance observed along the primate ventral visual stream, we subsequently propose *data augmentation invariance*, a simple, yet effective and efficient mechanism to improve the robustness of the representations: we include an additional, unsupervised term in the objective function that encourages the similarity between augmented examples within each batch.

In sum, the main contributions of this paper are the following:

- We propose a data augmentation invariance score that intuitively evaluates the robustness of the intermediate features of a neural network towards transformations of the input images.

- We use this score to show that the intermediate representations of several popular architectures do not become more disentangled than in the input space, after training.

- We propose an unsupervised, layer-wise learning objective that encourages the representational similarity between transformed examples.

## 2  METHODS

This section presents the procedure to empirically measure the robustness of the representations of a convolutional neural network and our proposal to improve the invariance.

### 2.1  EVALUATION OF INVARIANCE

To measure the invariance of the learned features under the influence of identity-preserving image transformations we compare the activations of a given image with the activations of a data augmented version of the same image.

Consider the activations of an input image $x$ at layer $l$ of a neural network, which can be described by a function $f^{(l)}(x) \in \mathbb{R}^{D^{(l)}}$. We can define the distance between the activations of two input images $x_i$ and $x_j$ by their mean square difference:

$$d^{(l)}(x_i, x_j) = \frac{1}{D^{(l)}} \sum_{k=1}^{D^{(l)}} (f_k^{(l)}(x_i) - f_k^{(l)}(x_j))^2 \tag{1}$$

Following this, we compute the mean square difference between $f^{(l)}(x_i)$ and a random transformation of $x_i$, that is $d^{(l)}(x_i, G(x_i))$. $G(x)$ refers to the stochastic function that transforms the input images according to a pre-defined data augmentation scheme.

In order to assess the similarity between the activations of an image $x_i$ and of its augmented versions $G(x_i)$ we normalize it by the average similarity in the (test) set. We define the invariance score $S_i^{(l)}$ of the transformation $G(x_i)$ at layer $l$ of a model, with respect to a data set of size $N$, as follows:

$$S_i^{(l)} = 1 - \frac{d^{(l)}(x_i, G(x_i))}{\frac{1}{N} \sum_{j=1}^{N} d^{(l)}(x_i, x_j)} \tag{2}$$

Note that the invariance $S_i^{(l)}$ takes the maximum value of 1 if the activations of $x_i$ and its transformed version $G(x_i)$ are identical.

## 2.2 DATA AUGMENTATION INVARIANCE

Most DNNs trained for object categorization are optimized through mini-batch gradient descent (SGD), that is the weights are updated iteratively by computing the loss of a batch $\mathcal{B}$ of examples, instead of the whole data set at once. The models are typically trained for a number of *epochs*, $E$, which is a whole pass through the entire training data set of size $N$. That is, the weights are updated $K = \frac{N}{|\mathcal{B}|}$ times each epoch.

Data augmentation introduces variability into the process by performing a different, stochastic transformation of the data every time an example is fed into the network. However, with standard data augmentation, the model has no information about the *identity* of the images, that is, that different augmented examples, seen at different epochs, separated by $\frac{N}{|\mathcal{B}|}$ iterations on average, correspond to the same seed data point. This information is potentially valuable and useful to learn better representations. For example, in biological vision, the high temporal correlation of the stimuli that reach the visual cortex may play a crucial role in the creation of robust connections (Wyss et al., 2006). However, this is generally not exploited in supervised settings. In semi-supervised learning, where the focus is on learning from fewer labeled examples, data augmentation has been used as a source of variability together with dropout or random pooling, among others (Laine & Aila, 2016).

In order to make use of this information and improve the robustness, we first propose to perform *in-batch* data augmentation by constructing the batches with $M$ transformations of each example (see Hoffer et al. (2019) for a similar idea). Additionally, we propose to modify the loss function to include an additional term that accounts for the invariance of the feature maps across multiple image samples. Considering the difference between the activations at layer $l$ of two images, $d^{(l)}(x_i, x_j)$, defined in Equation 1, we define the data augmentation invariance loss at layer $l$ for a given batch $\mathcal{B}$ as follows:

$$\mathcal{L}_{inv}^{(l)} = \frac{\sum_k \frac{1}{|\mathcal{S}_k|^2} \sum_{x_i, x_j \in \mathcal{S}_k} d^{(l)}(x_i, x_j)}{\frac{1}{|\mathcal{B}|^2} \sum_{x_i, x_j \in \mathcal{B}} d^{(l)}(x_i, x_j)} \tag{3}$$

where $\mathcal{S}_k$ is the set of samples in the batch $\mathcal{B}$ that are augmented versions of the same seed sample $x_k$. This loss term intuitively represents the average difference of the activations between the sample pairs that correspond to the same source image, relative to the average difference of all pairs. A convenient property of this definition is that $\mathcal{L}_{inv}$ does not depend on the batch size nor the number of in-batch augmentations $M = |\mathcal{S}_k|$. Furthermore, it can be efficiently implemented using matrix operations.

Since both, certain representational invariance at $L$ layers of the network and high object recognition performance at the network output are desired, we define the total loss as follows:

$$\mathcal{L} = (1 - \alpha)\mathcal{L}_{obj} + \sum_{l=1}^{L} \alpha^{(l)} \mathcal{L}_{inv}^{(l)} \tag{4}$$

where $\sum_{l=1}^{L} \alpha^{(l)} = \alpha$ and $\mathcal{L}_{obj}$ is the loss associated with the object recognition objective, typically the cross-entropy between the object labels and the output of a softmax layer. All the results we report in this paper have been obtained by setting $\alpha = 0.1$ and distributing the coefficients across the layers according to an exponential law, such that $\alpha^{(l=L)} = 10\alpha^{(l=1)}$. This aims at simulating a probable response along the ventral visual stream, where higher regions are more invariant than the early visual cortex[1].

## 2.3 ARCHITECTURES AND DATA SETS

As test beds for our hypotheses and proposal we train three neural network architectures: all convolutional network, All-CNN-C (Springenberg et al., 2014); wide residual network, WRN-28-10 (Zagoruyko & Komodakis, 2016); and DenseNet-BC (Huang et al., 2017). All three architectures have been widely used in the deep learning literature and they have distinctive architectural characteristics: only convolutional layers, residual blocks and dense connectivity, respectively.

We train the three architectures on the highly benchmarked data set for object recognition CIFAR-10 (Krizhevsky & Hinton, 2009). Additionally, in order to test our proposal on higher resolution images and a larger number of classes, we also train All-CNN and WRN on the *tiny* ImageNet data set, a subset of ImageNet (Russakovsky et al., 2015) with 100,000 64x64 training images that belong to 200 classes.

All models are trained using a data augmentation scheme that consists in affine transformations, contrast adjustment and brightness adjustment (see the details in Appendix A)

For the models trained on CIFAR-10, the training hyperparameters (learning rate, number of epochs, etc.) are set as in the original papers, except that, following the recommendation by Hernández-García & König (2018) we do not use explicit regularization (weight decay and dropout) since comparable performance is obtained without them if data augmentation is used.

On tiny ImageNet, All-CNN has three additional layers and is trained for 150 epochs, with batch size 128 and initial learning rate 0.01 decayed by 0.1 at epochs 100 and 125. WRN is trained for 50 epochs, with batch size 32 and initial learning rate 0.01 decayed by 0.2 at epochs 30 and 40. Both architectures are trained with stochastic gradient descent with Nesterov momentum 0.9.

Note that the hyperparameters were fine tuned for training only with the standard categorical cross-entropy and with standard epoch-wise data augmentation. Therefore, they are likely suboptimal for our proposed data augmentation invariance. However, our aim is not achieving the best possible classification performance, but rather demonstrate the suitability of data augmentation invariance and analyze the learned representations.

The invariance loss defined in Equation 3 was computed after the ReLU activation of each convolutional layer for All-CNN; at the output of each residual block for WRN, and after the first convolution and the output of each dense block for DenseNet.

## 3 RESULTS

One of the contributions of this paper is to empirically test in how far convolutional neural networks produce invariant representations under the influence of identity-preserving transformations of the input images. Figures 2–4 show the invariance scores, as defined in Equation 2, across the network layers. Since we do not compute the invariance score at every single layer of the architectures, the

---

[1]It is beyond the scope of this paper to analyze the sensitivity of the hyperparameters $\alpha^{(l)}$, but we have not observed a significant impact in the classification performance by using other distributions.

numbering of the layers simply indicate increasing depth in the hierarchy (see Section 2.3 for the details).

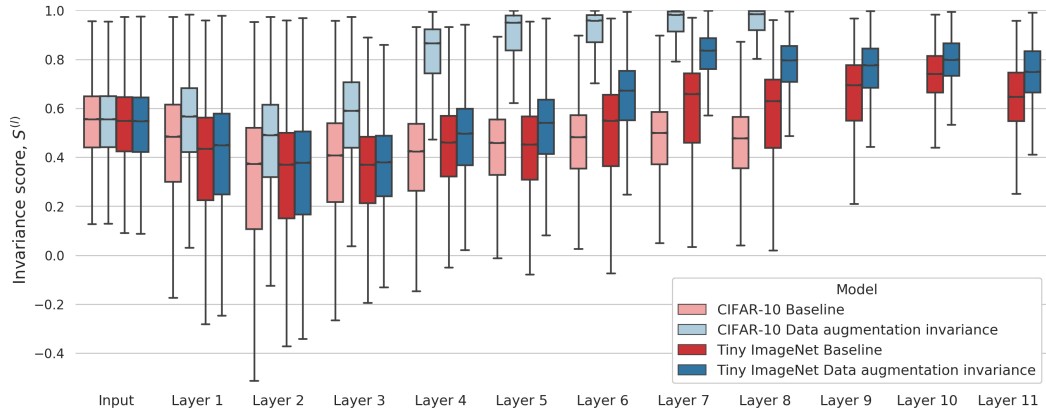

Figure 2: All-CNN: distribution of the invariance score at each layer of the baseline model and the model trained data augmentation invariance (higher is better).

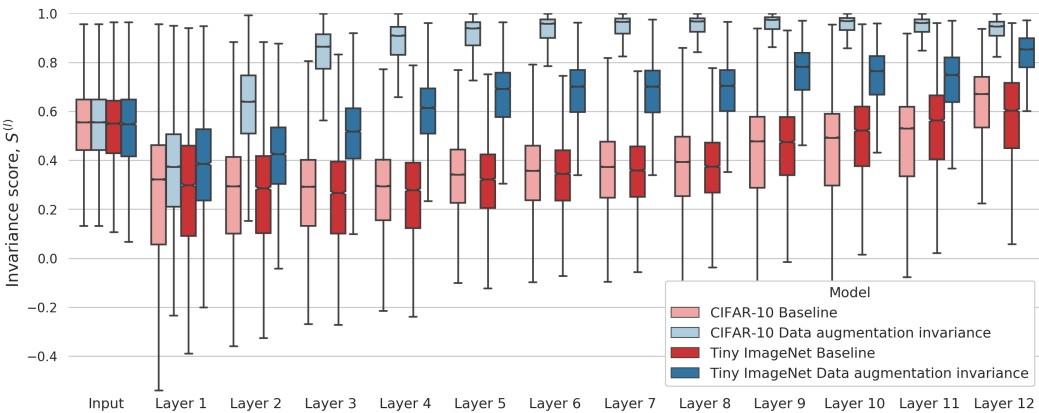

Figure 3: WRN: distribution of invariance score at each layer of the baseline model and the model trained data augmentation invariance (higher is better).

Despite the presence of data augmentation during training, which implies that the networks *see* and may learn augmentation-invariant transformations, the representations of the baseline models (red boxes) do not increase substantially beyond the invariance observed in the pixel space. As a solution, we have proposed a simple, unsupervised modification of the loss function to encourage the learning of data augmentation-invariant features. As can be seen in the plots (blue boxes), our data augmentation mechanism pushed network representations to become increasingly more robust with network depth.

Both All-CNN and WRN seem to more easily achieve the representational invariance on CIFAR-10 than on Tiny ImageNet. This may indicate that the complexity of the data set not only makes the object categorization more challenging, but also the learning of invariant features.

In order to better understand the effect of the data augmentation invariance, we plot the learning dynamics of the invariance loss at each layer of All-CNN trained on CIFAR-10. In Figure 5, we can see that in the baseline model, the invariance loss keeps increasing over the course of training. In contrast, when the loss is added to the optimization objective, the loss drops for all but the last layer. Perhaps unexpectedly, the invariance loss increased during the first epochs and only then started to decrease. While further investigations are required, these two phases may correspond to the compression and diffusion phases proposed by Shwartz-Ziv & Tishby (2017).

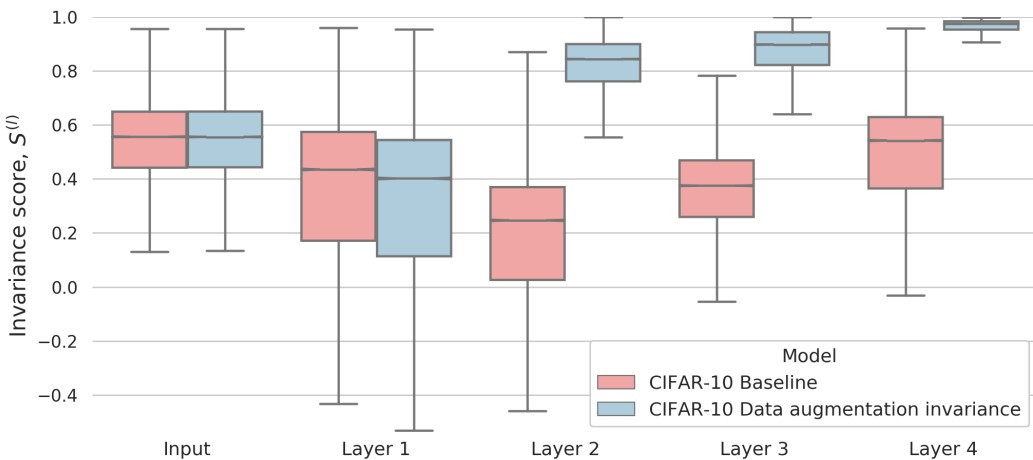

Figure 4: DenseNet: distribution of invariance score at each layer of the baseline model and the model trained data augmentation invariance (higher is better).

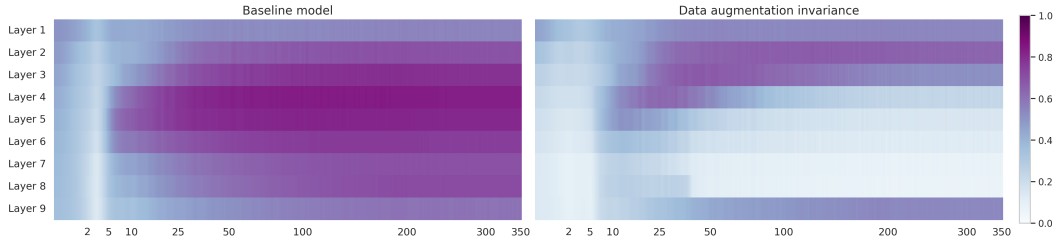

Figure 5: Dynamics of the data augmentation invariance loss $\mathcal{L}_{inv}^{(l)}$ during training (lower is better). The axis of abscissas (epochs) is scaled quadratically to better appreciate the dynamics at the first epochs. The same random initialization was used for both models.

In terms of efficiency, adding terms to the objective function implies an overhead of the computations. However, since the pairwise distances can be efficiently computed at each batch through matrix operations, the training time is only increased by about 10 %.

Finally, as reported in Table 1, the improved invariance comes at little or no cost in the categorization performance, as the networks trained with data augmentation invariance achieve similar classification performance to the baseline model. However, note that the hyperparameters used in all cases were optimized to maximize performance in the original models, trained without data augmentation invariance. Therefore, it is reasonable to expect an improvement in the classification performance if e.g. the batch size or the learning rate schedule are better tuned for this new learning objective. Learning increasingly invariant features could lead to more robust categorization, as exemplified by the increased test performance observed for the All-CNN models (despite no hyperparameter tuning).

Table 1: Classification accuracy on the test set of the baseline models and the models trained with data augmentation invariance.

|  | CIFAR-10 | | | Tiny ImageNet (acc. \| top5) | |
| --- | --- | --- | --- | --- | --- |
|  | All-CNN | WRN | DenseNet | All-CNN | WRN |
| Baseline | 91.48 | 94.58 | 94.88 | 51.09 \| 73.48 | 61.49 \| 82.99 |
| Data aug. invariance | 92.47 | 94.86 | 93.98 | 52.57 \| 76.53 | 61.23 \| 83.23 |

## 4 CONCLUSIONS

In this work, we have initially proposed an invariance score that assesses the robustness of the features learned by a neural network towards the identity-preserving transformations typical of common data augmentation schemes (see Equation 2). Intuitively, the more similar the representations of transformations of the same image, with respect to other images in a data set, the higher the data augmentation invariance score

Using this score, we have analyzed the features learned by three popular network architectures (All-CNN, WRN and DenseNet) trained on image object recognition tasks (CIFAR-10 and Tiny ImageNet). The analysis revealed that their features are less invariant than commonly expected, despite sufficient exposure to matching image transformations during training. In some cases, the representational invariance did not even increase with respect to the original pixel space. This property is fundamentally different to the primate ventral visual stream, where neural populations have been found to be increasingly robust to changes in view or lighting conditions of the same object (DiCarlo & Cox, 2007).

Taking inspiration from this property of the visual cortex, we have proposed an unsupervised objective to encourage learning more robust features, using data augmentation as the framework to perform identity-preserving transformations on the input data. We created mini-batches with $M$ augmented versions of each image and modified the loss function to maximize the similarity between the activations of the same seed images, as compared to other images in the training set. Aiming to approximate the observations in the biological visual system, higher layers were set to achieve exponentially more invariance than the early layers. Future work will investigate whether this increased robustness also allows for better modelling of neural data.

Data augmentation invariance effectively produced more robust representations, unlike standard models optimized only for object categorization, at little or no cost in classification performance and with an affordable, slight increase (10 %) in training time. Ideally, object recognition models should be reasonably invariant to all the transformations of the objects to which human perception is also invariant. Data augmentation is just an approximation to analyze and encourage invariance to a set of transformations that can be applied on still, 2D images. Future work should analyze the invariance of models trained with video and even 3D data.

These results provide additional empirical evidence that deep supervised models optimized only according to the standard categorization objective, that is the categorical cross-entropy between the true object labels and the model predictions, are able to successfully generalize to a held out test data set by learning discriminative, but non-robust features. This is likely to be due to their large capacity to learn discriminative features in a too unconstrained setting, which has been recently suggested to be at the source of adversarial vulnerability (Ilyas et al., 2019).

Finally, we have contributed to the growing body of evidence indicating that inspiration from biological vision can provide useful constraints for deep learning.

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

## A   DATA AUGMENTATION SCHEME

As specified in Section 2.3, the data augmentation scheme used to trained all the models consists of affine transformations, contrast adjustment and brightness adjustment. Specifically, we apply the following image transformations, with the parameters defined in Table 2:

- Affine transformations:
$$\begin{bmatrix} x' \\ y' \\ 1 \end{bmatrix} = \begin{bmatrix} f_h z_x \cos(\theta) & -z_y \sin(\theta + \phi) & t_x \\ z_x \sin(\theta) & z_y \cos(\theta + \phi) & t_y \\ 0 & 0 & 1 \end{bmatrix} \begin{bmatrix} x \\ y \\ 1 \end{bmatrix}$$
- Contrast adjustment: $x' = \gamma(x - \overline{x}) + \overline{x}$
- Brightness adjustment: $x' = x + \delta$

For the computation of the invariance score we use exactly the same transformations, but instead of randomly sampling for the parameter ranges defined in Table 2, we halve the range and sample from one of the extreme values.

Table 2: Description and range of possible values of the parameters used for the data augmentation scheme. $B(p)$ denotes a Bernoulli distribution and $\mathcal{N}(a, b)$ a truncated normal distribution centered at $\frac{a+b}{2}$ and with standard deviation $\frac{b-a}{4}$.

| Parameter | Description | Range |
|-----------|-------------|-------|
| $f_h$ | Horiz. flip | $1 - 2B(0.5)$ |
| $t_x$ | Horiz. translation | $\mathcal{N}(-0.1, 0.1)$ |
| $t_y$ | Vert. translation | $\mathcal{N}(-0.1, 0.1)$ |
| $z_x$ | Horiz. scale | $\mathcal{N}(0.85, 1.15)$ |
| $z_y$ | Vert. scale | $\mathcal{N}(0.85, 1.15)$ |
| $\theta$ | Rotation angle | $\mathcal{N}(-22.5°, 22.5°)$ |
| $\phi$ | Shear angle | $\mathcal{N}(-0.15, 0.15)$ |
| $\gamma$ | Contrast | $\mathcal{N}(0.5, 1.5)$ |
| $\delta$ | Brightness | $\mathcal{N}(-0.25, 0.25)$ |

