# OpenReview forum: "Learning robust visual representations using data augmentation invariance"
_ICLR.cc/2020/Conference — Reject_

### Official Review · AnonReviewer3 · 2019-10-23
**Official Blind Review #3**

**Rating:** 3

**Review:**

Motivated by biological visual systems, this paper investigates whether the representations of convolutional networks for visual recognition are invariant to identity preserving transformations. The results show that empirically they are not, and they further propose a data-augmentation approach to learn this invariance. Since transformations can be automatically generated, this does not require additional manual supervision.

The main weakness of this paper is that the approach is mostly data-augmentation, which is standard. Whereas standard data augmentation simply adds the transformed examples to the training set, this paper goes one step further and adds an additional regularization between two different views, so that they lie in the same space, such as using Euclidean or cosine distance. However, these loss functions are widely known and applied as a baseline in metric learning.

The problem itself is also well known in computer vision, and there are several empirical papers that demonstrate this. For example, the recent paper "Strike (with) a Pose: Neural Networks Are Easily Fooled by Strange Poses of Familiar Objects" at CVPR 2019.

Taking a step back, do we even want convolutional networks to be rotation invariant? The categorical label of some objects could change depending on the rotation. For example, a door is only a door if it is upright against a wall. If you rotate the door, it just becomes a board.

In my view, the novelty of this paper lies in the application of standard approaches to a standard vision problem. Due to this, I feel this contribution is not sufficient for ICLR.


**Experience Assessment:**

I have published one or two papers in this area.

**Review Assessment: Checking Correctness Of Derivations And Theory:**

I assessed the sensibility of the derivations and theory.

**Review Assessment: Checking Correctness Of Experiments:**

I assessed the sensibility of the experiments.

**Review Assessment: Thoroughness In Paper Reading:**

I read the paper thoroughly.

---

> ### Author Response · Authors · 2019-11-15
> **Is simplicity a weakness?**
>
> We first thank the reviewer for their feedback. Several interesting points were raised that we are happy to discuss next:
>
> According to the reviewer, the main weakness of the paper is that our approach makes use of a standard method (data augmentation) and a well-known distance metric (Euclidean). We humbly argue that rather than a weakness, the simplicity of our approach ought to be considered a strength. It would have been more questionable, for instance, if we had used a far-fetched distance metric to demonstrate our hypothesis. We use data augmentation as a framework precisely because it is a well-known method commonly used to transform existing images into new, plausible examples. Our work first demonstrates that the standard way of applying data augmentation yields representations with undesirable properties (lack of robustness), misaligned with a fundamental property of biological vision. Second, we show that a simple modification of the objective function can greatly improve the features robustness while preserving the classification performance. Simplicity is a positive aspect in this case, as it will be straightforward to incorporate the proposed method into existing models of image object classification.
>
> The reviewer also states that the problem of DNNs identified in our paper is well known in computer vision and points out to a recent paper. We would like to note that the problem we address in our work is fundamentally different to the one in the mentioned paper and other related works. While previous work reveals that DNNs fail at *classifying* objects on a different pose or perceptually similar (as in the problem of adversarial examples), we instead focus on the analysis of the intermediate features and reveal that DNNs represent transformations of the same image (obtained via standard data augmentation) very differently, even if they are correctly classified. To the best of our knowledge, this is a novel contribution.
>
> Reviewer: "Taking a step back, do we even want convolutional networks to be rotation invariant? The categorical label of some objects could change depending on the rotation. For example, a door is only a door if it is upright against a wall. If you rotate the door, it just becomes a board."
>
> In our opinion, we do want convolutional networks to be reasonably invariant to rotations of the objects within the ranges in which they are perceived (by humans) as the same objects in the real world. This is why we take inspiration from visual perception for our work and why we use data augmentation as a comprehensible way to generate "identity-preserving transformations". As illustrated by the reviewer, an extreme rotation of a door would change its category. In the particular case of rotation, humans likely perceive objects invariably within a range of rotation equivalent to the range in which the head can be tilted sideways. We aim at simulating such perceptual properties by setting appropriate data augmentation parameters. In particular, our data augmentation scheme performs rotations in the range of -22.5 to 22.5 degrees, sampled via a truncated normal distribution centered at 0 degrees. Therefore, our data augmentation scheme would never rotate a door image such that it would become a board. The parameters of the data augmentation scheme are detailed in the Appendix A.
>
> We hope this addresses the reviewer's concerns and we are open to further discussion.

---

### Official Review · AnonReviewer1 · 2019-10-27
**Official Blind Review #1**

**Rating:** 6

**Review:**

The paper proposes to explicitly improve the robustness of image-classification models to invariant transformations, via a secondary multi-task objective. The idea is that the secondary objective makes intermediate-layer representations invariant to transformations of the image that should lead to the same classification. The paper also establishes that the typical models do not actually learn such representations by themselves.

This is an interesting idea (I am not specialist enough on image classification to know whether this idea has been tried before.) and highly relevant to this conference. The paper is g. well-written, easy to read, and correct.

I do, however, rate it as Weak Accept only for one reason: I would expect that making the model more robust should improve classification accuracy. But according to the paper, accuracy does not improve (and even degrades slightly). The paper does not experimentally demonstrate that the proposed methods objectively improves the model.

In a sense, it is to be expected, since the model is no longer optimized 100% for the classification task.

I can think of three changes to the paper that would flip my review to a Strong Accept:

* Try using the multi-task objective as a pre-training, followed by fine-tuning without multi-task objective. This should foster robust internal representations while allowing to fully optimize for the classification task. Alternatively, you could anneal alpha to 0. Try whether this alleviates the losses on the tests that got worse, and leads to higher gains on the others.
* Maybe the robust models, while being worse on benchmarks, are better on real-life data, e.g. where training and test mismatches are higher. Can you find a test set that demonstrates a larger, reliable accuracy improvement from robustness?
* "However, note that the hyperparameters used in all cases were optimized to maximize performance in the original models, trained without data augmentation invariance. Therefore, it is reasonable to expect an improvement in the classification performance if e.g. the batch size or the learning rate schedule are better tuned for this new learning objective." -- Then that should be done.

Besides this, I have a few detailed feedback points:

Eq. (2): Using sigma for "invariance", which is the opposite of the usual meaning of sigma... I wish you had used a different symbol. Not a dealbreaker, but if you have a chance, it would be great to change to a different one.

"we normalize it by the average similarity with respect to the *other* images in the (test) set" -- If you use only *other* images, I think it is theoretically possible that sigma becomes negative. I think you should include the numerator image in the denominator as well. I understand that in practical terms, this is never going to be a problem, so it should be OK, no need to rerun anything.

"we first propose to perform in-batch data augmentation" -- This increases correlation of samples within the batch, and may therefore affect convergence. To be sure that this is not the cause of the degradation, it would be good to rerun the baseline with the same in-batch augmentation (but without the additional loss). Was that already done?

Figure 5: I was a little confused at first because I read this as the invariance score (like Figures 3 and 4), not the invariance loss. They seem to be opposite of each other. So I wondered why the right panel would show poorer invariance score as training progresses. Do you actually need the concept of "invariance score" at all? Or can you redefine the invariance score as a non-invariance score (1- of it), so that its polarity is the same as the invariance loss? If not, an easy fix would be to add "(lower=better)" to the caption of Fig. 5, and likewise ("higher=better") to Fig 3 and 4.

"DATA AUGMENTATION INVARIANCE" -- You really want more. You want to be robust to all valid transformations of the object in the images. Obviously it is not possible to augment data for that, but it would be good to state this somewhere as the idealized goal of this work, which you approximate by data augmentation.

**Experience Assessment:**

I have read many papers in this area.

**Review Assessment: Checking Correctness Of Derivations And Theory:**

I carefully checked the derivations and theory.

**Review Assessment: Checking Correctness Of Experiments:**

I carefully checked the experiments.

**Review Assessment: Thoroughness In Paper Reading:**

I read the paper thoroughly.

---

> ### Author Response · Authors · 2019-11-15
> **Some new results and feedback incorporated**
>
> We first sincerely thank the reviewer for their feedback. We especially appreciate the interesting suggestions.
>
> "I do, however, rate it as Weak Accept only for one reason: I would expect that making the model more robust should improve classification accuracy. But according to the paper, accuracy does not improve (and even degrades slightly). The paper does not experimentally demonstrate that the proposed methods objectively improves the model."
>
> We have a few comments in this regard:
>
> 1) We have 5 models to compare in Table 1 and only in two cases the accuracy slightly degraded: on DenseNet on CIFAR by -0.9; on WRN on Tiny ImageNet by -0.26 (but the top5 accuracy improved!) . In the rest of the cases, the accuracy improved: All-CNN on CIFAR +0.99; WRN on CIFAR by +0.28; All-CNN on Tiny ImageNet by +1.48 (and top5 by 3.05!).
>
> 2) The goal of the method was not to improve the accuracy, but increase the robustness of the features. As pointed out by the reviewer, "since the model is no longer optimized 100% for the classification task", it came as a surpirse that in some cases the performance even improved.
>
> 3) As also pointed by the reviewer, the hyperparameters are likely suboptimal because they were tuned for different conditions.
>
> The reviewer suggests a few improvements and experiments that would improve their rate of the paper. Due to time constraints and computational limitaitons (2 GPUs only), we could not try all the ideas on all architectures and data sets, but we have observed promising results. These are the experiments and results we obtained so far:
>
> - Data augmentation invariance pre-training (5 % of the total epochs) and subsequent annealing of alpha (suggested by the reviewer): there is a moderate improvement in the final classification accuracy:
> * All-CNN on CIFAR-10: 92.47 --> 92.87
> * WRN on CIFAR-10: 94.86 --> 95.17
>
> - Adjustment of the learning rate (divided by M=8):
> * All-CNN on CIFAR-10: 92.47 --> 93.11
> * WRN on CIFAR-10: 94.86 --> 95.47
>
> It seems that both ideas slightly increase the performance, especially adjusting the learning rate, since performing M in-batch data augmentation has an implicit effect of reducing the batch size by a factor of M (only approximately, since the augmented samples are only similar and not identical), and in turn, approximately equivalent to multiplying the learning rate by M. Pre-training and annealing alpha towards zero also seems to help, although it requires tuning several additional hyperparameters (epochs and decay factor). We will update the manuscript with the complete set of results, in case of acceptance.
>
> Regarding the rest of the feedback points:
>
> - We agreed on the issue about alpha. We have changed alpha by S. Thanks.
>
> - Denominator in Eq. 2. The actual operation is as indicated by the reviewer, that is we divide by the *total* average in the set. It was wrongly phrased in the paper and we have updated it.
>
> - "it would be good to rerun the baseline with the same in-batch augmentation": this was already the case. The baseline results are on the original models, with standard data augmentation.
>
> - We have added (higher is better) and (lower is better) to the captions accordingly. We preferred to keep the concept of invariance score, which is a desirable objective, which can be optimized by defining a loss, inverse of the invariance. This is analogous to classification accuracy and cross-entropy loss.
>
> Finally, we totally agree with the last remark. Data augmentation is, of course, just an approximation. We have added a paragraph discussion this in the last section of the paper.
>
> We are confident the feedback has improved our paper and hope the reviewer's concerns have been effectively addressed.

---

### Official Review · AnonReviewer4 · 2019-11-04
**Official Blind Review #4**

**Rating:** 3

**Review:**

This paper introduces an unsupervised learning objective that attempts to improve the robustness of the learnt representations. This approach is empirically demonstrated on cifar10 and tiny imagenet with different network architectures including all convolutional net, wide residual net and dense net.

This paper is well written and organized. However, I have several concerns.  The novelty of the proposed method is limited. The unsupervised objective in eq. (3) is a good and straightforward  engineering trick, but it is of less scientific interest.  The empirical results are not encouraging. Table1 shows the comparison results between the proposed method and a baseline method, however, for ALL-CNN, the reported top1 in the original paper is 92.75%. The reviewer is aware that this paper mentions the proposed method doesn't apply regularization, but why not compare to the original results. Results shown in figs. 2,3,4 are obvious because the proposed objective eq.3 prefers a higher invariance score in eq(2) and the increasing alphas prefer increasing invariance score.

In my opinion this work is not sufficient for ICLR.

**Experience Assessment:**

I have read many papers in this area.

**Review Assessment: Checking Correctness Of Derivations And Theory:**

I carefully checked the derivations and theory.

**Review Assessment: Checking Correctness Of Experiments:**

I assessed the sensibility of the experiments.

**Review Assessment: Thoroughness In Paper Reading:**

I read the paper at least twice and used my best judgement in assessing the paper.

---

> ### Author Response · Authors · 2019-11-15
> **On the novelty and the relevance of the results**
>
> First, we thank the reviewer for the assessment of our paper. We appreciate that some of the strengths have been identified.
>
> One concern is the novelty of the proposed method. In this regard, we would like to highlight that our paper makes two significant contributions: 1) we show that the features of DNNs (3 distinct, popular architectures) are surprisingly non-robust to the transformations of standard data augmentation. Note that, unlike many previous studies which have revealed that DNNs fail at *classifying* images which are perceptually similar (adversarial examples, noise [1], change of pose [2]), here we focus on the intermediate features and reveal that, even when similar images are classified correctly, their internal representations are remarkably different. To the best of our knowledge this is a novel contribution. 2) We propose a simple modification of the loss function that effectively and efficiently solves this robustness problem, while preserving or even improving the classification performance.
>
> According to the reviewer, our proposed solution is just a "straightforward  engineering trick, but it is of less scientific interest". This is surpising to us, especially because the proposal is motivated by visual neuroscience and perception. In particular, we aim at incorporating one of the key mechanisms for robust visual object categorization in the visual cortex, that is the invariance to identity-preserving transformations. The paper offers an introduction to this idea in the paper and reviews some of the relevant neuroscientific literature in this regard. This is rarely found in machine learning papers, which often focus indeed on engineering tricks with no scientific motivation.
>
> "Results shown in figs. 2,3,4 are obvious because the proposed objective eq.3 prefers a higher invariance score in eq(2) and the increasing alphas prefer increasing invariance score."
>
> This statement might be missing a subtle, but very important aspect of the results: all the results presented in our paper are obtained on test data, including the results in Figures 2, 3 and 4. According to the reviewer, it is "obvious" that the models obtain a higher invariance score, but this implies taking for granted good generalization given optimization on training data. This is equivalent to saying that it is obvious that models correctly classify objects in unseen images, while it is well-known that this is not the case in, for instance, challenging data sets such as ImageNet or the aforementioned problems of DNNs such as adversarial vulnerability. In other words, we humbly argue that it should not be taken for granted that optimizing robustness to identity-preserving transformation on the training data, should automatically grant robustness to *potentially different* transformations on unseen images.
>
> The reviewer further considers that the "empirical results are not encouraging", referring to the results presented in Table 1. We would first like to highlight that the main results of our paper are the ones that refer to the robustness of the features, which is the contribution of our work, in Figures 2, 3 and 4. The results in Table 1 aim to demonstrate that the modification of the objective function we propose does not significantly degrade the classification performance. Note that the motivation for our work is not to improve classification but to improve robustness. This is similar to works on adversarial robustness, which aim at reducing adversarial vulnerability, although many methods do impact the classification. Note that, nonetheless, our proposed method even improves the classification in some cases, surprisingly.
>
> Finally, following the reviewer's suggestion, we have carried out additional experiments including explicit regularization in the models. Unfortunately, due to time constraints and limited computational resources, we could only carry out a few pilot tests with subsets of data. The results we have observed are perfectly consistent with the results of the models trained without explicit regularization, therefore regularization seems to play no relevant role in this regard. We will add an appendix section to the manuscript including these results once they are available. The reason why we trained without explicit regularization is that it has been shown that it is unnecessary if data augmentation is applied [3] and their hyperparameters are extremely sensitive to changes in the learning procedure, as is our case by changing the objective function and applying in-batch data augmentation.
>
> We hope this discussion effectively addressed the reviewer's concerns.
>
> [1] Geirhos et al.. Generalisation in humans and deep neural networks. NeurIPS, 2018.
>
> [2] Alcorn et al.. Strike (with) a Pose: Neural Networks Are Easily Fooled by Strange Poses of Familiar Objects. CVPR, 2019.
>
> [3] Hernández-García and König. Data augmentation instead of explicit regularization. arXiv preprint arXiv:1806.03852, 2018.

---

### Decision · Program_Chairs · 2019-12-19

**Decision:**

Reject

**Comment:**

This paper introduces an unsupervised learning objective that attempts to improve the robustness of the learnt representations. This approach is empirically demonstrated on cifar10 and tiny imagenet with different network architectures including all convolutional net, wide residual net and dense net.   Two of three reviewers felt that the paper was not suitable for publication at ICLR in its current form.  Self supervision based on preserving network outputs despite data transformations is a relatively minor contribution, the framing of the approach as inspired by biological vision notwithstanding.  Several references, including at a past ICLR include:
http://openaccess.thecvf.com/content_CVPR_2019/papers/Kolesnikov_Revisiting_Self-Supervised_Visual_Representation_Learning_CVPR_2019_paper.pdf
and
Gidaris, P. Singh, and N. Komodakis. Unsupervised representation learning by predicting image rotations.  In International Conference on Learning Representations (ICLR), 2018.